# Skeletal muscle loss and body composition in progressive supranuclear palsy: A retrospective cross-sectional study

Yasuyuki Takamatsu[1], Ikuko Aiba[2]*

1 Faculty of Health Sciences, Department of Rehabilitation Science, Hokkaido University, Sapporo, Japan,
2 Department of Neurology, National Hospital Organization, Higashinagoya National Hospital, Nagoya, Japan

* ikukoaiba0401@gmail.com

## Abstract

### Introduction

Skeletal muscle mass loss has been associated with decreased physical performance; however, the body composition characteristics in progressive supranuclear palsy (PSP) are not well understood. We investigated body composition parameters, focusing on skeletal muscle mass, in patients with PSP and compared them with those of healthy older adults.

### Methods

This retrospective cross-sectional study included 39 patients with PSP and 30 healthy older adults (control group). Using a multi-frequency bioelectrical impedance analysis, we measured the skeletal mass index (SMI), basal metabolism, extracellular water/total body water ratio (ECW/TBW), and body fat percentage and examined the relationship between SMI and age, body mass index (BMI) and other body composition parameters.

### Results

The PSP group had a higher rate of low muscle mass (56.4%) than the control group (10.0%), although the ages and BMIs were similar. The leg SMI was lower for the PSP group, while the ECW/TBW was higher for the PSP group. The basal metabolism was lower for the PSP group than for the controls but only in the women. The basal metabolism and BMI showed a significant correlation with SMI in the PSP group. There was a significant correlation between SMI and age, ECW/TBW, and body fat percentage in the PSP group but only in the women.

### Conclusion

This study is the first to show that a high proportion of patients with PSP have low muscle mass. We showed differences in terms of sex in muscle mass loss in women with PSP, which was associated with inactivity and aging.

**Data Availability Statement:** All relevant data are within the paper.

**Funding:** This work was supported by Grants-in Aid from the Research Committee of CNS Degenerative Diseases, Research on Policy

Planning and Evaluation for Rare and Intractable Diseases, Health, Labour and Welfare Sciences Research Grants, the Ministry of Health, Labour and Welfare, Japan (20FC1049, to I.A.). The funders had no role in study design, data collection and analysis, decision to publish, or preparation of the manuscript.

**Competing interests:** Aiba (corresponding author) serves as a consultant for Biogen MA Inc. and AbbVie GK and agree with the PLOS ONE policies on sharing data and materials.

## Introduction

Age-related loss of skeletal muscle mass and reduced muscle strength are collectively known as sarcopenia. In 2010, the European Working Group on Sarcopenia in Older People (EWGSOP) published a definition of sarcopenia that has been widely employed by clinicians and researchers [1]. Sarcopenia has been associated with motor functional disability, lower quality of life (QOL), and mortality [2–4]. In addition to aging, other factors related to the development of sarcopenia include immobility, malnutrition, and chronic disease [1]. Sarcopenia has also been reported in Parkinson's disease [5,6].

Progressive supranuclear palsy (PSP) is a neurodegenerative disorder that is considered an atypical parkinsonian syndrome [7,8]. PSP progresses faster than Parkinson's disease, and the clinical treatment for PSP has not been well-established, unlike Parkinson's disease. Patients with PSP exhibit supranuclear gaze palsy, falls, postural instability, gait disturbance, and cognitive impairment [7,8] and tend to be older. Their loss of physical function might therefore be due to both disease-related and age-related sarcopenia. Loss of skeletal muscle mass has been associated with a decline in physical performance and activities of daily living (ADL) [9]. However, no studies have examined body composition and skeletal muscle mass in PSP. We therefore investigated the body composition, with a focus on skeletal muscle mass, of patients with PSP compared with that of healthy older adults.

## Methods

### Participants

We conducted a retrospective cross-sectional study at the National Hospital Organization Higashinagoya National Hospital in Nagoya, Japan, between June 2017 and August 2020, which included patients with probable or possible PSP according to the 2017 Movement Disorder Society Criteria for the Clinical Diagnosis of PSP [8,10]. For comparison, we also enrolled age-matched neurologically healthy community-dwelling older adults into a control group. The PSP group consisted of inpatients hospitalized for rehabilitation or respite care, as well as outpatients. The exclusion criteria were as follows: 1) under 60 years of age, 2) requiring more than minor help with ADL, 3) a modified Rankin scale (mRS) [11] score $\geq$ 5, and 4) the inability to measure one's own body composition. The study was conducted in accordance with the principles of the Declaration of Helsinki and was approved by the Ethics Committee of the National Hospital Organization Higashinagoya National Hospital (approval number: 28–13, 30–11). The healthy older adults provided their written informed consent after receiving a verbal explanation of the study. The patients' informed consent was obtained in the form of an opt-out on the website.

### Assessments

The age, sex, height, weight, and body mass index (BMI) of all participants were recorded. We measured their skeletal muscle mass, basal metabolism, body fat percentage (BFP), and extracellular water to total body water ratio (ECW/TBW) with the InBody S10 body composition analyzer (InBody Japan, Inc., Tokyo, Japan), which employs a multi-frequency bioelectrical impedance analysis [12–14]. We calculated the skeletal muscle mass index (SMI) as follows: [appendicular skeletal muscle mass (kg)]/[height (m)]$^2$. Based on the Asian Working Group on Sarcopenia (AWGS) criteria for sarcopenia in older people [15], we defined low muscle mass as an SMI < 7.0 kg/m$^2$ in men and <5.7 kg/m$^2$ in women and investigated the number and proportion in each group for the main endpoint. Based on the World Health Organization's body fat thresholds for diagnosing obesity as recommended by the American

Association of Clinical Endocrinology [16], we defined obesity as a BFP >25% for men and >35% for women. We recorded the disease duration, mRS score, and Barthel index (BI) for the patients with PSP. All assessments of the inpatients with PSP were performed in the first few days after hospital admission to exclude the effects of treatment and rehabilitation.

## Statistical analysis

We performed the statistical analysis using SPSS software, version 26 (IBM, Inc., Armonk, NY, USA), which excluded missing values. All statistical comparisons were performed by sex, analyzing the number of male and female participants with low muscle mass (sex-specific SMI below the AWGS criteria) and obesity (sex-specific BFP above the WHO criteria). We performed Fisher's exact method to analyze the participants by low muscle mass, obesity, and sex, and calculated the effect size ($|\varphi|$). For the other assessments, we applied Mann-Whitney U test and calculated the effect size ($|r| = Z/\sqrt{n}$). We employed Spearman's rank-correlation coefficient to investigate the relationship between SMI and the other parameters in each group. A p-value < 0.05 was considered significant.

## Results

### Study patients

The study included 39 patients with PSP and 30 control participants, with no significant differences between the groups in terms of age (Table 1).

### Comparison of body composition between the controls and the patients

The SMI of the PSP group was significantly lower than that of the control group in both the men (p = 0.010) and women (p < 0.001). In particular, the mean leg SMI of the PSP group was significantly lower than that of the controls for the men (p = 0.001) and women (p < 0.001) but not the mean arm SMI. The rate of participants with low muscle mass was significantly higher in the PSP group (56.4%) than in the control group (p < 0.001), with a similar result when analyzing by sex (58.3% of men, p = 0.012; 53.3% of women, p = 0.002). In the women, the basal metabolism was significantly lower in the PSP group than in the control group (p < 0.001), but there were no significant differences in the men. The ECW/TBW was significantly higher in the PSP group than in the control group in the men (p = 0.004) and women (p = 0.003). There were no significant differences in BMI. BFP showed no significant differences, but the women in the PSP group tended to have a higher BFP than those in the control group. The frequency of obesity was significantly higher in the PSP group than in the control group (p = 0.042). In the men, there was no significant difference in the frequency of obesity, whereas in the women, the frequency of obesity was significantly higher in the PSP group than in the control group (p = 0.018).

### Correlation between the skeletal mass index and other factors in each group and by sex

The older women with PSP showed a lower SMI (p = 0.033), but there was no correlation between age and SMI in the other groups. There was a significantly positive correlation between BMI and SMI, except in the women in the control group (p = 0.009 for the women with PSP, p = 0.001 for the men in the control group, and p < 0.001 for the men with PSP). There was a moderate and higher positive correlation between basal metabolism and SMI in all groups, although there was no significant correlation for the women in the control group (p < 0.001 for the women with PSP, p < 0.001 for the men in the control group, and p < 0.001

**Table 1. Participants' demographic and clinical characteristics.**

| | Total | | | | Female | | | | Male | | | |
|---|---|---|---|---|---|---|---|---|---|---|---|---|
| | Controls | PSP | | | Controls | PSP | | | Controls | PSP | | |
| N | 30 | 39 | p | Effect size | 13 | 15 | p | Effect size | 17 | 24 | p | Effect size |
| Age, years [a] | 71.8 ± 6.3 | 73.8 ± 6 | – | – | 71.2 ± 7.8 | 72.1 ± 5.9 | 0.683 | 0.08 [d] | 72.2 ± 5 | 74.9 ± 5.9 | 0.151 | 0.22 [d] |
| SMI, kg/m² [a] | 7.3 ± 0.7 | 6.4 ± 1.1 | – | – | 6.9 ± 0.5 | 5.5 ± 1.0 | <0.001** | 0.70 [d] | 7.6 ± 0.7 | 7.0 ± 0.8 | 0.010* | 0.41 [d] |
| Arm SMI, kg/m² [a] | 1.6 ± 0.4 | 1.5 ± 0.3 | – | – | 1.4 ± 0.2 | 1.3 ± 0.3 | 0.316 | 0.20 [d] | 1.7 ± 0.4 | 1.7 ± 0.2 | 0.711 | 0.06 [d] |
| Leg SMI, _kg/m² [a] | 5.8 ± 0.5 | 4.9 ± 0.8 | – | – | 5.6 ± 0.5 | 4.2 ± 0.7 | <0.001** | 0.74 [d] | 5.9 ± 0.4 | 5.3 ± 0.6 | 0.001** | 0.52 [e] |
| Low SMI frequency, % [b] | 3 (10) | 22 (56.4) | <0.001** | 0.48 | 0 (0) | 8 (53.3) | 0.002** | 0.59 [e] | 3 (17.6) | 14 (58.3) | 0.012* | 0.41 [d] |
| BM, Kcal/day [a] | 1338.3 ± 129.7 | 1244.6 ± 165.2 | – | – | 1253 ± 53.5 | 1089.7 ± 94.9 | <0.001** | 0.73 [d] | 1398.5 ± 134.7 | 1341.5 ± 119.0 | 0.255 | 0.18 [d] |
| ECW/TBW [a] | 0.383 ± 0.007 | 0.392 ± 0.010 | – | – | 0.384 ± 0.007 | 0.391 ± 0.007 | 0.003** | 0.56 [d] | 0.383 ± 0.007 | 0.393 ± 0.012 | 0.004** | 0.45 [d] |
| BMI, kg/m² [a] | 22.2 ± 2.6 | 21.5 ± 3.7 | – | – | 21.9 ± 2.0 | 21.6 ± 4.9 | 0.964 | 0.01 [d] | 22.4 ± 3.1 | 21.4 ± 2.9 | 0.169 | 0.21 [d] |
| BFP, % [a] | 21.3 ± 7 | 24.9 ± 10 | – | – | 23.3 ± 7.3 | 30.5 ± 11.7 | 0.088 | 0.33 [d] | 19.8 ± 6.6 | 21.5 ± 6.9 | 0.624 | 0.08 [d] |
| Obesity frequencies, % [b] | 3 (10) | 13 (33.3) | 0.042* | 0.27 [e] | 0 (0) | 6 (40) | 0.018* | 0.49 [e] | 3 (17.6) | 7 (29.2) | 0.480 | 0.132 [e] |
| Disease duration, years [a] | – | 4.6 ± 2.6 | – | – | – | 4.2 ± 2.0 | – | – | – | 4.8 ± 3.0 | – | – |
| mRS [c] | – | 4 [3–4] | – | – | – | 3 [3–4] | – | – | – | 4 [3–4] | – | – |
| Barthel index [a] | – | 49.2 ± 22.2 | – | – | – | 55.0 ± 25.8 | – | – | – | 45.7 ± 19.7 | – | – |

Controls, healthy older adult group; BFP, body fat percentage; BI, Barthel index; BM, basal metabolism; BMI, body mass Index; ECW, extracellular water; mRS, modified Rankin scale; PSP, progressive supranuclear palsy group; SMI, skeletal muscle mass index; TBW, total body water.

Results are reported as mean ± standard deviation [a], numbers (%) [b], and median [interquartile range] [c]. Effect size was calculated using the following formula; $|r| = Z/\sqrt{n}$ [d] and $|\varphi|$ [e].

* indicates $p < 0.05$

** indicates $p < 0.01$.

for the men with PSP). There was significant correlation between ECW/TBW and SMI for the women with PSP (p = 0.011) but not for the other groups. There was a moderately significant correlation between BFP and SMI for the women with PSP (p = 0.009) but not for the other groups. In the entire PSP group, there was no significant correlation between SMI on one hand and disease duration, mRS and BI on the other (Table 2).

## Discussion

This study is the first to investigate body composition in PSP, focusing on the skeletal muscle mass of patients with PSP in detail and comparing it to that of healthy older adults. We found that the PSP group had a greater proportion of individuals with low muscle mass compared with the group of healthy older participants.

The SMI was significantly lower in the PSP group than in the healthy control group, and many of the patients with PSP met the AWGS criteria for sarcopenia, although the BMI was similar. Our findings are of value in clarifying the characteristics of PSP by measuring body composition. In addition, there was no significant difference in the participants' age between

**Table 2. Spearman's rank-correlation coefficient between skeletal muscle mass index and the other parameters in each group.**

| CON Females | SMI | Age | BMI | BM | ECW/TBW | BFP | | | |
|---|---|---|---|---|---|---|---|---|---|
| SMI | 1.000 | −0.121 | 0.138 | 0.524 | −0.091 | −0.044 | | | |
| Age | - | 1.000 | 0.273 | −0.800** | −0.148 | 0.257 | | | |
| BMI | - | - | 1.000 | −0.053 | 0.077 | 0.784** | | | |
| BM | - | - | - | 1.000 | −0.118 | −0.014 | | | |
| ECW/TBW | - | - | - | - | 1.000 | 0.035 | | | |
| BFP | - | - | - | - | - | 1.000 | | | |
| CON Males | SMI | Age | BMI | BM | ECW/TBW | BFP | | | |
| SMI | 1.000 | −0.380 | 0.711** | 0.897** | 0.025 | 0.049 | | | |
| Age | - | 1.000 | −0.206 | −0.326 | 0.023 | −0.245 | | | |
| BMI | - | - | 1.000 | 0.505* | 0.068 | 0.574* | | | |
| BM | - | - | - | 1.000 | 0.057 | −0.218 | | | |
| ECW/TBW | - | - | - | - | 1.000 | 0.140 | | | |
| BFP | - | - | - | - | - | 1.000 | | | |
| PSP Females | SMI | Age | BMI | BM | ECW/TBW | BFP | Duration | mRS | BI |
| SMI | 1.000 | −0.553* | 0.925** | 0.939** | −0.636* | 0.650** | −0.116 | −0.293 | 0.365 |
| Age | - | 1.000 | −0.378 | −0.706** | 0.589* | −0.070 | 0.203 | −0.086 | −0.232 |
| BMI | - | - | 1.000 | 0.843** | −0.621* | 0.793** | −0.115 | −0.409 | 0.471 |
| BM | - | - | - | 1.000 | −0.600* | 0.496 | −0.211 | −0.177 | 0.272 |
| ECW/TBW | - | - | - | - | 1.000 | −0.393 | −0.204 | 0.239 | −0.417 |
| BFP | - | - | - | - | - | 1.000 | 0.142 | −0.571* | 0.516* |
| Duration | - | - | - | - | - | - | 1.000 | −0.031 | −0.144 |
| mRS | - | - | - | - | - | - | - | 1.000 | −0.839** |
| BI | - | - | - | - | - | - | - | - | 1.000 |
| PSP Males | SMI | Age | BMI | BM | ECW/TBW | BFP | Duration | mRS | BI |
| SMI | 1.000 | 0.018 | 0.663** | 0.886** | −0.032 | 0.096 | 0.209 | −0.137 | 0.205 |
| Age | - | 1.000 | 0.093 | 0.079 | 0.527** | 0.177 | 0.039 | 0.414* | −0.214 |
| BMI | - | - | 1.000 | 0.539** | −0.138 | 0.742** | 0.307 | −0.020 | 0.231 |
| BM | - | - | - | 1.000 | −0.020 | 0.022 | 0.215 | −0.128 | 0.208 |
| ECW/TBW | - | - | - | - | 1.000 | −0.159 | 0.190 | 0.163 | 0-.453* |
| BFP | - | - | - | - | - | 1.000 | 0.154 | −0.004 | 0.211 |
| Duration | - | - | - | - | - | - | 1.000 | 0.014 | −0.065 |
| mRS | - | - | - | - | - | - | - | 1.000 | −0.682** |
| BI | - | - | - | - | - | - | - | - | 1.000 |

BFP, body fat percentage; BI, Barthel index; BMI, body mass Index; BM, basal metabolism; CON, healthy older adult controls; ECW, extracellular water; mRS, modified Rankin Scale; PSP, progressive supranuclear palsy group; SMI, skeletal muscle mass index; TBW, total body water.

* indicates p < 0.05

** indicates p < 0.01.

the patients with PSP and the healthy participants in this study. Therefore, skeletal muscle mass loss in PSP (as with other neurodegenerative diseases) might be due to inactivity, which is known to induce muscle fiber atrophy [17]. The main pathological lesions in PSP occur in the substantia nigra, tegmentum, pallidum, subthalamic nucleus, and cerebellum [7,18]. Therefore, patients with PSP exhibit bradykinesia, axial rigidity, and gait disturbance [7,8], which result in inactivity. Patients with PSP in this study showed low ADL levels, as shown by mRS and BI.

The increase in ECW/TBW showed edema, which was higher in the PSP group than in the healthy control group. Furthermore, the rate of obesity among the women with PSP, who tended to have higher BFP, was greater than among the healthy women. It has been reported that skeletal muscle loss can be accompanied by the accumulation of ectopic fat within muscles and has been associated with reduced motor function [19]. Our findings therefore show a change in body composition in PSP, with a reduction in skeletal muscle mass [16].

We found differences in terms of sex in the factors related to skeletal muscle mass loss in PSP, the first of which was the relationship between SMI and age. There was no significant correlation between SMI and age in the men with PSP. Their skeletal muscle mass loss was therefore likely induced by the disease rather than by aging [17]. However, the women with PSP showed a significant correlation between SMI and age, which indicates that the skeletal muscle mass loss could be induced not only by inactivity but also by aging in the women with PSP, unlike the men with PSP. The second point is the relationship between SMI and basal metabolism. Skeletal muscle mass was significantly related to basal metabolism, both in the men and women. However, the basal metabolism showed a significant correlation with age in the women but not the men. Furthermore, the women with PSP showed a greater effect size of skeletal muscle mass loss compared with the men with PSP. Women with PSP might show significantly lower basal metabolism compared with healthy women due to a reduction in skeletal muscle mass with aging and inactivity.

Our study had several limitations, the first of which was the first of which was its small sample size, which prevented us from performing a multivariate regression to show the causal relationship between the factors. The second limitation was that this study was lack of data about related factors and confounders causing skeletal muscle mass loss or sarcopenia, such as motor function (e.g., muscle strength, gait speed) [1,15], daily physical activity (e.g., exercise state) [20,21], nutrition sate (e.g., caloric pattern, intake of ergogenic or ergogenic drugs, bulbar dysfunction, anorexia) [22–25]. Lastly, this was a retrospective cross-sectional single-center study. A future longitudinal multicenter study is needed to investigate the effects of low skeletal muscle mass on the functional prognosis in PSP.

In conclusion, we investigated the body composition parameters in patients with PSP, focusing on skeletal muscle mass, and compared them with those of healthy older adults. We showed that the PSP group had a high rate of low muscle mass compared with the healthy group. We also showed differences in terms of sex in muscle mass loss in the women with PSP, which was associated not only with inactivity but also with aging.

## Acknowledgments

We would like to thank the patients and their families for their contributions. We would also like to thank Dr. Satoko Sakakibara (Department of Neurology), Dr. Misaki Sato (Department of Neurology), Naomi Matsuda, and the other physical therapists (Department of Rehabilitation) at the National Hospital Organization Higashinagoya National Hospital for their support.

## Author Contributions

**Conceptualization:** Yasuyuki Takamatsu, Ikuko Aiba.

**Data curation:** Yasuyuki Takamatsu, Ikuko Aiba.

**Formal analysis:** Yasuyuki Takamatsu.

**Funding acquisition:** Ikuko Aiba.

**Investigation:** Ikuko Aiba.

**Methodology:** Yasuyuki Takamatsu, Ikuko Aiba.

**Project administration:** Yasuyuki Takamatsu, Ikuko Aiba.

**Supervision:** Ikuko Aiba.

**Validation:** Yasuyuki Takamatsu, Ikuko Aiba.

**Visualization:** Yasuyuki Takamatsu.

**Writing – original draft:** Yasuyuki Takamatsu.

**Writing – review & editing:** Ikuko Aiba.

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
