## [Decision Letter · Decision Letter 0]

16 Apr 2021

PONE-D-21-02875

Skeletal muscle loss and body composition in progressive supranuclear palsy: A retrospective cross-sectional study

PLOS ONE

Dear Dr. Aiba,

Thank you for submitting your manuscript to PLOS ONE. After careful consideration, we feel that it has merit but does not fully meet PLOS ONE’s publication criteria as it currently stands. Therefore, we invite you to submit a revised version of the manuscript that addresses the points raised during the review process.

We look forward to receiving your revised manuscript.

Kind regards,

Ezio Lanza, M.D.

Academic Editor

PLOS ONE

**Additional Editor Comments **:

Please consider citing in the introduction the following article, published  by PLOS ONE,  regarding the use of sarcopenia as predictor of survival also in the setting of cancer  

https://journals.plos.org/plosone/article/comments?id=10.1371/journal.pone.0232371

Journal Requirements:

"I.A. serves as a consultant for Biogen MA Inc. and AbbVie GK."

3. We note you have included a table to which you do not refer in the text of your manuscript. Please ensure that you refer to Table 2 in your text; if accepted, production will need this reference to link the reader to the Table.

Reviewers' comments:

Reviewer's Responses to Questions

5. Review Comments to the Author

Reviewer #1: I thank the authors for taking consideration of the relation between body composition parts and PSP patients. In order for the manuscript to be scientifically valid regarding the association suggested, I suggest that adjustment for other confounders causing wasting and sarcopenia should be mentioned in the manuscript [exercise state of the patient versus controls, the caloric pattern [moderate, mild or high according to daily activity and occupation], details of ergogenic or ergopenic medications and nutritional state including the presence or absence of bulbar dysfunction or anorexia]. These factors should be elucidated in my experience in order to make the publication worthy of targeting the association between PSP and sarcopenia.

Reviewer #2: Manuscript by Takamatsu and Aiba is the first to report skeletal muscle loss in progressive supra nuclear palsy (PSP). Small size of studied groups of patients and controls is a significant limitation. However comparative analyses performed by sex, identified significant decrease in SMI in PSP groups driven by leg SMI.

This paper should be of interest in the field of clinical PSP.

---

## [Author Response · Author response to Decision Letter 0]

22 Apr 2021

To editor and reviewers

We thank the editor and reviewers for taking the time to review our manuscript. We have revised the manuscript according to the editor’s and reviewers’ comments. The revised portions of the manuscript are highlighted in red.

Additional Editor Comments:

Please consider citing in the introduction the following article, published by PLOS ONE, regarding the use of sarcopenia as predictor of survival also in the setting of cancer 

https://journals.plos.org/plosone/article/comments?id=10.1371/journal.pone.0232371

Following the editor’s suggestion, we cited the following article in the introduction: Lanza E et al. Sarcopenia as a predictor of survival in patients undergoing bland transarterial embolization for unresectable hepatocellular carcinoma. PLoS One. 2020.

Introduction: page 3, lines 44-45.

Sarcopenia has been associated with motor functional disability, lower quality of life (QOL), and mortality [2–4].

References: page 11, lines 233-236.

4. Lanza E, Masetti C, Messana G, Muglia R, Pugliese N, Ceriani R, et al. Sarcopenia as a predictor of survival in patients undergoing bland transarterial embolization for unresectable hepatocellular carcinoma. PLoS One. 2020;15: 1–12. doi:10.1371/journal.pone.0232371

 

Journal Requirements:

We confirmed our reference list to be complete and correct.

Following the journal requirements, we revised the formatting of our manuscript based on these sample PDFs.

"I.A. serves as a consultant for Biogen MA Inc. and AbbVie GK."

Yes, we agree with the PLOS ONE policies on sharing data and materials. We mentioned about COI in cover letter.

3. We note you have included a table to which you do not refer in the text of your manuscript. Please ensure that you refer to Table 2 in your text; if accepted, production will need this reference to link the reader to the Table.

Following the journal requirements, we revised the manuscript.

Results: page 7, line 144.

In the entire PSP group, there was no significant correlation between SMI on one hand and disease duration, mRS and BI on the other (Table 2).

 

Reviewers' comments:

Reviewer #1: I thank the authors for taking consideration of the relation between body composition parts and PSP patients. In order for the manuscript to be scientifically valid regarding the association suggested, I suggest that adjustment for other confounders causing wasting and sarcopenia should be mentioned in the manuscript [exercise state of the patient versus controls, the caloric pattern [moderate, mild or high according to daily activity and occupation], details of ergogenic or ergopenic medications and nutritional state including the presence or absence of bulbar dysfunction or anorexia]. These factors should be elucidated in my experience in order to make the publication worthy of targeting the association between PSP and sarcopenia.

Thank you for raising very important points. The remark about confounders causing sarcopenia is critical and the lack of data was a significant limitation in this study. Therefore, following the reviewer’s comment, we revised the manuscript and added some references.

Discussion: page 9, lines 185-189.

The second limitation was that this study was lack of data about related factors and confounders causing skeletal muscle mass loss or sarcopenia, such as motor function (e.g., muscle strength, gait speed) [1,15], daily physical activity (e.g., exercise state) [20,21], nutrition sate (e.g., caloric pattern, intake of ergogenic or ergogenic drugs, bulbar dysfunction, anorexia) [22–25]. 

References: page 13, line 294 – page 14, line 312.

20. Steffl M, Bohannon R w, Sontakova L, Tufano JJ, Shiells K, Holmerova I. Relationship between sarcopenia and physical activity in older people: a systematic review and meta-analysis. Clin Interv Aging. 2017;Volume 12: 835–845. doi:10.2147/CIA.S132940

21. Lee SY, Tung HH, Liu CY, Chen LK. Physical Activity and Sarcopenia in the Geriatric Population: A Systematic Review. J Am Med Dir Assoc. 2018;19: 378–383. doi:10.1016/j.jamda.2018.02.003

22. Naseeb MA, Volpe SL. Protein and exercise in the prevention of sarcopenia and aging. Nutr Res. 2017;40: 1–20. doi:10.1016/j.nutres.2017.01.001

23. Cruz-Jentoft AJ, Kiesswetter E, Drey M, Sieber CC. Nutrition, frailty, and sarcopenia. Aging Clin Exp Res. 2017;29: 43–48. doi:10.1007/s40520-016-0709-0

24. Azzolino D, Damanti S, Bertagnoli L, Lucchi T, Cesari M. Sarcopenia and swallowing disorders in older people. Aging Clin Exp Res. 2019;31: 799–805. doi:10.1007/s40520-019-01128-3

25. Zhao WT, Yang M, Wu HM, Yang L, Zhang X mei, Huang Y. Systematic Review and Meta-Analysis of the Association Between Sarcopenia and Dysphagia. J Nutr Heal Aging. 2018;22: 1003–1009. doi:10.1007/s12603-018-1055-z

 

Reviewer #2: Manuscript by Takamatsu and Aiba is the first to report skeletal muscle loss in progressive supra nuclear palsy (PSP). Small size of studied groups of patients and controls is a significant limitation. However comparative analyses performed by sex, identified significant decrease in SMI in PSP groups driven by leg SMI.

This paper should be of interest in the field of clinical PSP.

Thank you for reviewing our manuscript and being interest. We would like to increase the sample size and proceed with further verification.

---

## [Decision Letter · Decision Letter 1]

28 May 2021

Skeletal muscle loss and body composition in progressive supranuclear palsy: A retrospective cross-sectional study

PONE-D-21-02875R1

Dear Dr. Aiba,

We’re pleased to inform you that your manuscript has been judged scientifically suitable for publication and will be formally accepted for publication once it meets all outstanding technical requirements.

Kind regards,

Ezio Lanza, M.D.

Academic Editor

PLOS ONE

Reviewer #1: All comments have been addressed

Reviewer #2: All comments have been addressed

---

## [Editor Report · Acceptance letter]

2 Jun 2021

PONE-D-21-02875R1 

Skeletal muscle loss and body composition in progressive supranuclear palsy: A retrospective cross-sectional study 

Dear Dr. Aiba:

I'm pleased to inform you that your manuscript has been deemed suitable for publication in PLOS ONE. Congratulations! Your manuscript is now with our production department. 

Kind regards, 

on behalf of

Dr. Ezio Lanza 

Academic Editor

PLOS ONE